



# Modeling soil organic carbon dynamics and its driving factors in global main cereal cropping systems

Guocheng Wang[1*], Wen Zhang[1*], Wenjuan Sun[2], Tingting Li[1], Pengfei Han[1]

[1]*State Key Laboratory of Atmospheric Boundary Layer Physics and Atmospheric Chemistry, Institute of Atmospheric*

*Physics, Chinese Academy of Sciences, Beijing, China*

[2]*State Key Laboratory of Vegetation and Environmental Change, Institute of Botany, Chinese Academy of*

*Sciences, Beijing 100093, China*

*Correspondence to*: Guocheng Wang (wanggc@mail.iap.ac.cn)

**Abstract**

The net fluxes of carbon dioxide ($CO_2$) between the atmosphere and agricultural systems are mainly characterized by

the changes in soil carbon stock, which is determined by the balance between carbon input from organic materials and

output through soil C decomposition. The spatiotemporal changes of cropland soil organic carbon (SOC) in response

to different carbon (C) input management and environmental conditions across the global main cereal systems were

studied using a modeling approach. We also identified the key variables driving SOC changes at a high spatial

resolution ($0.1° \times 0.1°$) and long time scale (54 years from 1961 to 2014). The widely used soil C turnover model

(RothC) and the state-of-the-art databases of soil and climate were used in the present study. The model simulations

suggested that, on a global average, the cropland SOC density increased at an annual rate of 0.22, 0.45 and 0.69 MgC

ha$^{-1}$ yr$^{-1}$ under a crop residue retention rate of 30%, 60% and 90%, respectively. Increased quantity of C input could

enhance the soil C sequestration or reduce the soil C loss rate, depending largely on the local soil and climate

conditions. Spatially, under a certain crop residue retention rate, a relatively higher soil C sink were generally found

across the central parts of the United States, western Europe, northern regions of China, while a relatively smaller soil

C sink generally occurred in regions at high latitudes of both northern and southern hemisphere, and SOC decreased across the equatorial zones of Asia, Africa and America. We found that SOC change was significantly influenced by the crop residue retention rate (linearly positive), and the edaphic variable of initial SOC content (linearly negative). Temperature had weakly negative effects, and precipitation had significantly negative impacts on SOC changes. The results can help target carbon input management for effectively mitigating climate change through cropland soil C sequestration on a global scale.

## 1 Introduction

Soil is the largest terrestrial carbon (C) pool, and a small variation in soil carbon stock can lead to substantial changes in atmospheric carbon dioxide ($CO_2$) concentrations (Schlesinger and Andrews, 2000;Scharlemann et al., 2014). Soil organic carbon (SOC) stored in croplands constitute around 10% of the global soil carbon stock (Jobbagy and Jackson, 2000), and cultivation generally leads to marked changes in SOC through influencing the processes regarding soil C production and decomposition (Luo et al., 2013;Wang et al., 2016). Cropland SOC changes are regulated by complex interactions between the local soil environmental and climatic conditions, as well as the management regimes (Brady and Weil, 2008). Moreover, there lacks a continuity in soil C monitoring data over meaningfully large scales of both time and space. Consequently, the ability to characterize the SOC dynamics on a fine spatiotemporal resolution over a large scale is substantially hindered.

Basically, cropland SOC is a balance of carbon inputs (mainly dependant on biomass productivity that controlled by the climate and management conditions) and outputs (strongly regulated by climatic conditions). Since the start of 1960s, the "green revolution" aiming at providing more food to feed the increasing population has been widely launched across the global agricultural systems (Evenson and Gollin, 2003). During this period, numerous efforts regarding the crop variety improvement, and application of water irrigation and nitrogen fertilization have been taken to enhance the global crop production (Fischer and Edmeades, 2010;Evenson and Gollin, 2003). As a result, the global crop production tripled from 1961 to 2010, which is due mostly to greater yield per unit area (Zeng et al., 2014).



Increases in crop production would certainly provide more carbon inputs (e.g., organic matters from crop roots and residues) into soils, thereby substantially affecting the SOC sequestration (Wang et al., 2016). However, such impacts at fine spatiotemporal resolutions on a global scale is still unclear and has seldom been comprehensively studied.

During the past several decades, a number of agricultural system models have been developed and used to reproduce the dynamic processes including carbon flows between the agro-ecosystems and the atmosphere (Li et al., 1994;Parton et al., 1994;Keating et al., 2003;Huang et al., 2009). These models have been reported to be able to capture the soil C changes under different environmental and management conditions, thereby providing an opportunity for quantifying soil C dynamics at larger scales over time and space. Based on the process-based models, efforts have already been taken to quantify the national and continental scale cropland soil C dynamics. For example, using the Century model, Ogle et al. (2010) and Lugato et al. (2014), respectively, estimated that the average cropland soil C density increased at a rate of 1.3 Mg C ha$^{-1}$ yr$^{-1}$ from 1990 to 2000 in US, and 0.12 Mg C ha$^{-1}$ yr$^{-1}$ from 2013 to 2050 in Europe under improved management. Using another biogeophysical model (i.e., Agro-C), Yu et al. (2012) quantified that China's cropland soils have been annually sequestering around 0.20 Mg C ha$^{-1}$ from 1980 to 2009. Using the same model, however, Wang et al. (2013) found that the average soil C annually decreased 0.20 Mg C ha$^{-1}$ from 1960 to 2010 in the Australian wheat-belt. The large disparities in either the sign or the magnitude in soil C changes could be attributed to the different local soil and climate conditions and/or agricultural management practices. Moreover, differences in regional model input data obtained from different sources and/or simulating procedures such as model configurations and parameterizations in different studies with different models can also bias the regional simulation results, thereby hampering a comprehensive and robust evaluation of cropland soil C dynamics on a global scale.

Currently, most existing process-based models require many detailed parameters as the model inputs, which were not readily obtainable on a large scale. As one of the most classic and widely used soil C turnover models, the RothC model (Jenkinson et al., 1990), however, requires only a few and easily obtainable parameters and input data. The model has already been widely and frequently adopted to simulate the soil C changes under different management

treatments and soil and climate conditions across the world's cropping systems (Falloon and Smith, 2002;Guo et al., 2007;Yang et al., 2003;Bhattacharyya et al., 2011;Skjemstad et al., 2004;Smith et al., 2005). More recently, by adopting the model's original default parameters, the RothC has been tested against the measurements obtained from 16 long-term experimental sites across the global croplands, and showed a general good performance in representing

the SOC dynamics under different treatments at different sites (Wang et al., 2016).

In this study, we simulated the spatiotemporal soil C dynamics across the global main cereal (i.e., wheat, maize and rice) cropping systems, using the RothC model and state-of-the-art databases of soil and climate. The soil C revolutions were simulated under different scenarios of C inputs (calculated from crop residues, roots and manure) on a monthly time step from 1961 to 2014, at a high spatial resolution of $0.1° \times 0.1°$. Based on the model simulations, we

presented the spatiotemporal changes in SOC across the global main cereal growing areas under different residue retention rates. The impacts of C input management, edaphic and climatic variables on SOC changes were also statistically analyzed to identify the key factors driving the soil C dynamics.

## 2 Materials and methods

*Study area*

The study area covered the main cereal (i.e., wheat, maize and rice) cropping regions in the world (Fig. S1). We selected the wheat, maize and rice cropping areas because they are the most widely planted (covering around 72% of global cereal cropping areas) and productive (constituting around 80% of global cereal yield) cereals in the world (FAOSTAT, 2017). The geographic distribution of the global croplands ($0.1° \times 0.1°$ spatial resolution, with a cropland percentage value within each pixel) (Ramankutty et al., 2008), and the growing areas of wheat, maize and rice

(Monfreda et al., 2008) were sourced from the Center for Sustainability and the Global Environment (SAGE). The main cereal cropping regions were then obtained through masking the global croplands by cropping areas of wheat, maize and rice using GIS analysis approach. According to Vancutsem et al. (2013), we selected the pixels with more



than 30% cropland areas as the study area in the present study (Fig. S1), this is because such pixels can in general more efficiently represent the croplands in the real world.

*RothC model and its initialization*

The Rothamsted carbon model (RothC, version 26.3) was used to simulate the cropland soil C dynamics in the present
study. RothC is a widely used soil organic matter (SOM) decomposition model to simulate C dynamics in agricultural soils under various environments and management (Smith et al., 2005;Guo et al., 2007;Skjemstad et al., 2004). Recently, Wang et al. (2016) evaluated the model's performance in simulating soil C variations using observations of 16 long-term experimental sites across the world's wheat-growing regions. The validating results suggested that the model could reasonably reproduce the SOC dynamics under a wide range of soil and climatic conditions and
agricultural management practices. Detailed information of the RothC model description can be found in Jenkinson et al. (1990).

Soil carbon pool in RothC model is divided into five conceptual components, i.e., decomposable plant material (*DPM*), resistant plant material (*RPM*), microbial biomass (*BIO*), humified organic matter (*HUM*), and inert organic matter (*IOM*). These conceptual pools can hardly be directly measured in most cases and can only be empirically initialized
because only the quantity of total soil organic carbon is obtainable without finer level partitioning among the sub-pools. In the present study, following Wang et al. (2016), we adopted the approach of Weihermüller et al. (2013), who developed a validated set of pedotransfer functions to initialize C pools in the RothC model:

$$IOM = 0.049 \times SOC^{1.139} \tag{1}$$

$$RPM = (0.1847 \times SOC + 0.1555) \times (Clay + 1.2750)^{-0.1158} \tag{2}$$

$$HUM = (0.7148 \times SOC + 0.5069) \times (Clay + 10.3421)^{0.0184} \tag{3}$$

$$BIO = (0.0140 \times SOC + 0.0075) \times (Clay + 8.8473)^{0.0567} \tag{4}$$

where *SOC* is the total soil organic C content in the top 30 cm soil layer (Mg C ha$^{-1}$) and *Clay* is the soil clay fraction

(%).

The default yearly decomposition rate for the above-mentioned five soil C sub-pools were divided by 12 in order to run the model on a monthly time step (Jenkinson et al., 1990). The annual carbon input from crop residue, root and manure to soils were assumed to occur at the time after harvests, which is acceptable because the model is insensitive

to the time of C input, particularly in long-term simulations (Smith et al., 2005). The default value of DPM/RPM ratio (i.e., 1.44) of the C input is adopted in this study because it is suggested as a typical value for most agricultural crops (Jenkinson et al., 1990).

*Spatial data*

Soil parameters used in the present study such as soil carbon density and clay fraction in the top 30 cm soil profiles

were sourced from the Harmonized World Soil Database (Fao and Isric, 2012). This soil dataset combines information from various sources such as WISE, SOTER and FAO Soil Map of the world, and it is recommended as the most recent and most detailed globally consistent and continuous map of SOC with a highest spatial resolution of $0.1° \times$ $0.1°$ (Fig. S2) so far available (Hiederer and Köchy, 2011;Scharlemann et al., 2014). The soil cover information were derived from the crop calendar dataset (Sacks et al., 2010), which were documented in the Center for Sustainability

and the Global Environment (SAGE) and provides gridded maps of global crop planting and harvesting dates for 19 major crops including wheat, maize and rice.

The global climate data layers at $0.5° \times 0.5°$ spatial resolution (Harris et al., 2014) were sourced from the Climatic Research Unit (https://crudata.uea.ac.uk/cru/data/hrg/). The most recent version of climate data product (i.e., CRU TS v.4.00) was used in this study. The monthly time-series climate data layers include mean air temperature, precipitation

and potential evapotranspiration, spanning from 1901 to 2014. According to Jenkinson et al. (1990), the potential evapotranspiration was converted to open pan evaporation (one of the required model inputs of RothC) by dividing 0.75, i.e., open pan evaporation = potential evapotranspiration / 0.75. The climate data has a coarser spatial resolution





than that of the soil dataset (i.e., 0.1°× 0.1°), at which we performed the RothC model simulations. Here, the climate data in each coarser pixel were assumed to be the same as that in the finer pixels (0.1°× 0.1°) locate within that coarser pixel (0.5°× 0.5°).

The carbon inputs are mainly sourced from crop residues, roots and manure (Yu et al., 2012). We derived these information on a high spatial resolution from various sources of existing datasets. Firstly, the crop yield for wheat, maize and rice in 2005 on a global scale at a 0.1°× 0.1° spatial resolution were obtained from the Spatial Production Allocation Model (SPAM) 2005 (You et al., 2014). The SPAM provides crop-specific information on yield at a high spatial resolution, and it has undergone a significant validation and has shown promising performance globally (Liu et al., 2010). However, the SPAM dataset does not include a continuous time-series data. As such, we adopted the global annual change rates of the major cereal crop yields at a 0.1°× 0.1° resolution (Ray et al., 2012) to generate a time-series crop yield data. Here, we calculated the annual crop yields from 1961 to 2014 based on the annual percent change rates of crop yield and the crop yield data in 2005 (i.e., SPAM dataset), by assuming a linear change rate in the crop yields. This is acceptable because the global yield increase rates have been found linear for most major cereal crops since the start of 1960s (Fischer and Edmeades, 2010;Hafner, 2003). In each grid, the annual amounts of crop residue and root were then calculated based on the yield data by adopting the residue/economic product ratio and root/shoot ratio as described by Huang et al. (2007). All residues and roots were assumed to have a carbon content of 45% in determining the quantity of carbon input from crops (Skjemstad et al., 2004).

Annual carbon input contributed by manure application at a global scale were derived from Zhang et al. (2017), who used the dataset from Global Livestock Impact Mapping System (GLIMS) in conjunction with country-specific annual livestock population to reconstruct the manure nitrogen production and application in global croplands during 1860-2014 at a high spatial resolution of 0.1°× 0.1°. Following Lugato et al. (2014), the C input to soils from manure was calculated according to the average C:N ratio of different type of manures. The average C:N ratio of manures was



set to 20 because various studies have found that manure in general maintains a relatively stable C:N ratio of around 20 (Sharpley and Moyer, 2000;Ko et al., 2008;Eghball et al., 2002). The calculated C inputs from crop roots and manure were assumed to be all incorporated into soils. The amount of C inputs from crop above-ground residues, however, were further determined by setting different residue retention scenarios as described below.

5    *Scenario simulations and identifying controls on SOC dynamics*

The above-ground residue retention rates show a vast spatial disparity across the global croplands, and generally increase from 30% in the developing regions such as Asia and Africa (Jiang et al., 2012;Baudron et al., 2014) to more than 60% in the developed regions such as Europe and North America (Scarlat et al., 2010;Lokupitiya et al., 2012). To the best of our knowledge, there lacks a detailed information on residue retention rate over a meaningfully large scale of both time and space across different countries and continents. Consequently, a scenario modeling approach was adopted. In general, we specified three crop residue retention rates in the present study, i.e., 30%, 60% and 90% (hereafter simply denoted as R30, R60 and R90, respectively).

In total, we ran 461,586 (3 crop residue retention scenarios × 153,862 grids) RothC simulations. Each simulation quantified SOC content in the top 30 cm of soil from 1961 to 2014 on a monthly step. Based on the model simulations, we showed the spatiotemporal changes of SOC under different crop residue retention rates. We also assessed the impacts of crop residue retention, climatic and soil variables on SOC change using Spearman's rank correlation coefficient (*rho*). Selected climatic variables included mean annual temperature (hereafter simply denoted as *temperature*) and mean annual precipitation (hereafter simply denoted as *precipitation*). This is because these two variables have been suggested uncorrelated and could reasonably represent the spatial variation in a wide range of climate patterns (Bryan, 2003). For correlation analysis, the long-term monthly climate variables were summarized to mean annual values for each grid. Selected soil parameters included the model's edaphic inputs, i.e., initial SOC content and soil clay fraction. Change in soil C is calculated as the difference in SOC between 2014 and 1961. Spearman's rank correlation coefficient was then calculated between SOC change and crop residue retention rates and

soil and climate variables across the full set of RothC simulations. The sign of *rho*, positive or negative, indicates the direction of association between the independent and dependent variables. The absolute magnitude of *rho*, between 0 and 1, suggests the strength of correlation between the two variables. All analyses were performed using statistical and graphical software R 3.3.2 (R Development Core Team, 2017).

## 3 Results

On a global average, soil organic carbon (SOC) generally increased over time under different specified crop residue retention rates in the present study (Fig. 1). The median SOC increased from 46.2 MgC ha$^{-1}$ in 1961 to 58.3 MgC ha$^{-1}$ under R30 (Fig. 1a), and to 70.9 MgC ha$^{-1}$ under R60 (Fig. 1b), and to 84.1 MgC ha$^{-1}$ under R90 (Fig. 1c), respectively, in 2014. In general, the annual changing rates in SOC were 0.22 MgC ha$^{-1}$ yr$^{-1}$ under R30, 0.45 MgC ha$^{-1}$ yr$^{-1}$ under R60, and 0.69 MgC ha$^{-1}$ yr$^{-1}$ under R90, respectively.

Figure 2 shows the spatial patterns of the estimated SOC changes under R30 (Fig. 2a), R60 (Fig. 2b) and R90 (Fig. 2c), respectively. Among the three scenarios, a relatively higher increase in SOC generally occurred at the middle latitudes of the northern hemisphere such as the central parts of the United States, western Europe, northern regions of China (Fig. 2). A relatively small increase in SOC generally occurred in regions at high latitudes of both northern and southern hemisphere, while SOC decreased across the equatorial zones of Asia, Africa and America (Fig. 2). On a global average, 69%, 82% and 89% of the study area acted as a net carbon sink during the study period under R30 (Fig. 2a), R60 (Fig. 2b) and R90 (Fig. 2c), respectively.

The quantified SOC changes also showed large spatiotemporal disparities across different continents (Fig. 3). In generally, among the three scenarios, the cropland SOC across Europe, Asia, North America generally showed a linearly increasing trend over time (Fig. 3). In Oceania, SOC increased faster in the first two decades and showed a relatively lower increasing rate during the later three decades (Fig. 3). In South America and Africa, SOC decreased in the first few decades and increased or remained relatively stable during the later periods under R30 (Fig. 3a) and R60 (Fig. 3b). Under R90, however, the average SOC in all continents increased over time (Fig. 3c). In general, regions

with higher annual C input rates (e.g., Europe and North America) experienced higher SOC increases than those areas with relatively lower C input rates (e.g., Oceania and Africa), across all the three crop residue retention scenarios (Fig 3 and S4).

The quantified SOC changes were regulated by soil, climate and management practices. Initial SOC was significantly but negatively correlated ($rho$ = -0.20) with SOC change, while soil clay fraction showed a negligible correlation ($rho$ = -0.17, Fig. 4). The selected climatic variables displayed a negligible correlation (temperature, $rho$ = -0.18), and a significant but negative correlation (precipitation, $rho$ = -0.22) with SOC change, respectively (Fig. 4). Crop residue retention rate showed a strong and positive correlation ($rho$ = 0.34) with SOC change (Fig. 4). Figure 5 presented the impacts of crop residue retention, initial SOC content and precipitation, respectively, on SOC change. In general, crop residue retention seemed to linearly and positively correlate with SOC change (Fig. 5a), whereas initial SOC content (Fig. 5b) and precipitation (Fig. 5c) had a linearly negative effect on SOC change.

**4 Discussion**

*4.1 Interpretation and implication of the results*

Soil organic carbon change is a balance between C input from crops and manures and C output through decomposition. The quantified linear increase in global average SOC (Fig. 1) can mainly be attributed to the increased C input rate through the study period (Fig. S3). This is associated with the increased crop production since the start of "green revolution" launched during the 1960s (Fischer and Edmeades, 2010;Evenson and Gollin, 2003). In the present study, we found that the crop residue retention rate is strongly and positively correlated with SOC change (Fig. 4). This is similar to the findings of our previous studies (Wang et al., 2016;Wang et al., 2015), that higher amount of C input can lead to higher cropland soil C sink capacity. On a global average, enhancing the crop residue retention rate from 30% to 60% and 90% approximately induced a double and triple SOC sequestration rate, respectively (Fig. 1 and Fig. S3). However, it should be noted that the increased SOC sequestration rate contributed by increased C input amount can be limited at longer periods as the SOC would eventually reach a threshold of a relatively stable level (Stewart et al.,

2007).

Apart from the residue retention rate, the initial SOC is one of the major controlling factors on SOC change. The results in Figure 4 and 5 indicates that under otherwise similar environmental and managed conditions, soils with lower initial SOC content would experience greater SOC increases or smaller soil C losses. This negative correlation

between SOC change and initial SOC content has also been documented in other studies (Zhao et al., 2013;Wang et al., 2014). The relationship is further supported by the quantified distribution of global SOC change (Fig. 2) and the global initial SOC density (Fig. S2). For example, soils with lower initial SOC content in western Europe generally showed a higher SOC increase than that in the eastern Europe with relatively higher initial SOC content (Fig. 2 and Fig. S2). Such spatial patterns that lower initial SOC associated with higher SOC changes in neighboring areas can also be

found in other regions such as the United States and China (Fig. 2 and Fig. S2). Soil clay fraction has been suggested to benefit C stabilization through mineralogical protection of soil C (Oades, 1988;Amato and Ladd, 1992), whereas we identified a negligible but negative correction between soil C accumulation and soil clay fraction (Fig. 4). This adverse effects of soil clay could be a result of the strong correlation between initial soil C content and soil clay fraction ($rho$ = 0.31, data not shown). Here, soils with higher initial SOC content generally had a higher clay fraction, and this would

overshade the contributions of soil clay in benefiting soil C accumulation.

The negative effects of higher temperature and precipitation on SOC change identified in the present study (Fig.4 and 5) can be attributed to the higher SOC decomposition rate in warmer and wetter soils, which is consistent with the RothC model's description (Jenkinson et al., 1990) and the other findings of Bond-Lamberty and Thomson (2010). Here, it should be noted that such correlations between climate and SOC change might only be valid in a soil carbon turnover

model consisting only the C dynamic processes in the soil (e.g., RothC model). In other agricultural system model simulations, climatic variables may play a different role in affecting SOC change through jointly regulating both crop productions and soil C dynamics. For example, Wang et al. (2014) used a process-based agricultural system model (i.e., Agro-C model) to simulate the SOC dynamics in the semi-arid regions of North China Plain, and found a positive





effects of temperature and precipitation on SOC accumulation. This is because, in the temperature and water deficient areas (e.g., North China Plain), increased temperature and precipitation promoted crop production and hence increasing the C input to soils and favoring SOC sequestration.

Can we estimate the actual historical soil C dynamics across the world? There exists a big challenge due mainly to a lack of data availability, particularly for the two main RothC model inputs such as initial SOC content and annual C input. Firstly, the soil properties presented with the HWSD were derived from different sources with uneven sampled soil profiles over time and space. As such, the value of initial SOC content can hardly, if not impossible, represent the actual SOC content in the beginning of the study period. However, the modeled dynamics of SOC in the present study can be valid, to a certain extent, to represent the spatiotemporal patterns of the soil C source/sink processes. Second, there remains a lack of detailed information on crop residue management across both time and space, which also hinders our ability to accurately characterize the SOC changes on a large scale at fine spatiotemporal resolutions. Such as it is, we can still roughly assume that the above-ground residue retention rates were generally 30% in developing regions such as Asia and Africa (Jiang et al., 2012;Baudron et al., 2014;Erenstein, 2011), and 60% in other regions (Lokupitiya et al., 2012;Scarlat et al., 2010;Baudron et al., 2015). Based on these assumptions, we furthered quantified that the global average SOC increased at a rate of 0.34 MgC ha$^{-1}$ yr$^{-1}$ at an average annual C input rate of 2.4 MgC ha$^{-1}$ yr$^{-1}$ from 1961 to 2014. On a global scale, the estimated efficiency of conversion of input C to SOC (i.e., ratio of SOC change to C input) equals to 14%, which falls into the range of Campbell et al. (2000)'s result of 10-18%. It should be noted that the conversion efficiency varies across space and is highly dependent on local climatic and edaphic conditions (Yu et al., 2012).

By extrapolating these results to the world's whole croplands with a total area of 1,400 Mha (Jobbagy and Jackson, 2000), the global cropland soils could have annually sequestered 0.48 Pg C from 1961 to 2014, which equals to around 8% of the contemporaneous global average annual C emissions from fossil fuel combustions (http://cdiac.ornl.gov/ftp/ndp030/global.1751_2014.ems). Through enhancing the crop residue retention rate to 60%



and 90% in all the global croplands, soil C accumulation would offset around 11% and 16%, respectively, of the fossil fuel-induced C emissions. Again, it is noteworthy that although soil C can be increased by enhancing the quantity of C input, it would eventually reach a threshold at a higher level (Stewart et al., 2007). Until then, more amount of carbon input would be needed to maintain the soil C at higher levels (Wang et al., 2016). Otherwise, the cropland soil C would decrease and soils would act as a net C source again.

*4.2 Uncertainties and limitations*

Several uncertainties and limitations should be noticed in interpreting the simulation results in this study. First, the modeled SOC change in the present study could be biased due to the spatial inconsistency in time of soil sampling, which generally ranges widely during the second half of the twentieth century (Fao and Isric, 2012). In some places, the initial soil C information derived from HWSD could only represent the actual soil C levels during the later periods other than the early 1960s . For example, the soil profile measurements used for producing the soil map of China, which is further included in the HWSD datasets, were generally made in the late 1970s and early 1980s (Yu et al., 2007). Considering that the cropland SOC across space could have substantially changed over the study period under the changing environments and management practices (Fig 1 and 2), the initial SOC used in the present study (derived from HWSD) might significantly differ from the actual soil C levels in the early 1960s. Besides, it has been reported that soils with higher initial C content would experience smaller increase or greater C loss under otherwise similar conditions, and *vice versa* (Zhao et al., 2013;Wang et al., 2015). Consequently, for those regions with soil sampling time much later than early 1960s, our quantified SOC changes could be underestimated in the areas with substantial soil C increase had occurred before then. On the contrary, the SOC changes could be overestimated across the areas accompanying with a previous significantly decreased soil C.

Second, the RothC model was developed for simulating the soil organic matter turnover in upland soils (Jenkinson et al., 1990), and it generally showed well performance across the global wheat systems with non-waterlogged soils (Wang et al., 2016). In the paddy soils, particularly during the rice-growing seasons, the soil C decomposition rate

might be reduced subjected to anaerobic conditions (Shirato and Yokozawa, 2005). Consequently, the RothC model, used in the present study, could have underestimated the SOC changes across the rice systems that mainly distributed in the Southeast Asia.

Last but not least, the limitations of current first-order decay model (e.g., RothC) may cause significant bias in the model simulations. For example, our results suggested a general linear relation between C input and SOC variation (Fig. 1 and S3), which contradicts the findings of increasing the amount of crop residue incorporation may affect the SOC change in variable ways other than linearly (Powlson et al., 2011). Moreover, it has been reported that although soil can accumulate a significant amount of C when the pre-existing soil C content is low, SOC reaches a certain higher level (i.e., carbon saturation state) with little or no significant further change even with more C inputs (Stewart et al., 2007;Qin et al., 2013). Without considering the C saturation state, the first-order decay model might overestimate the SOC at longer time scale simulations particularly in regions with higher C input and lower SOC decomposition rate.

## Acknowledgements

This research was funded by the National Natural Science Foundation of China (Grant No. 41590870, 31370492).

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







**Figure 1.** Temporal changes in soil organic carbon (MgC ha$^{-1}$) of the global main cereal cropping regions under different above-ground crop residue retention rates of 30% (a), 60% (b) and 90% (c). Boxplots show the median and interquartile range, with whiskers extending to the most extreme data point within $1.5 \times (75\text{-}25\%)$ data range.


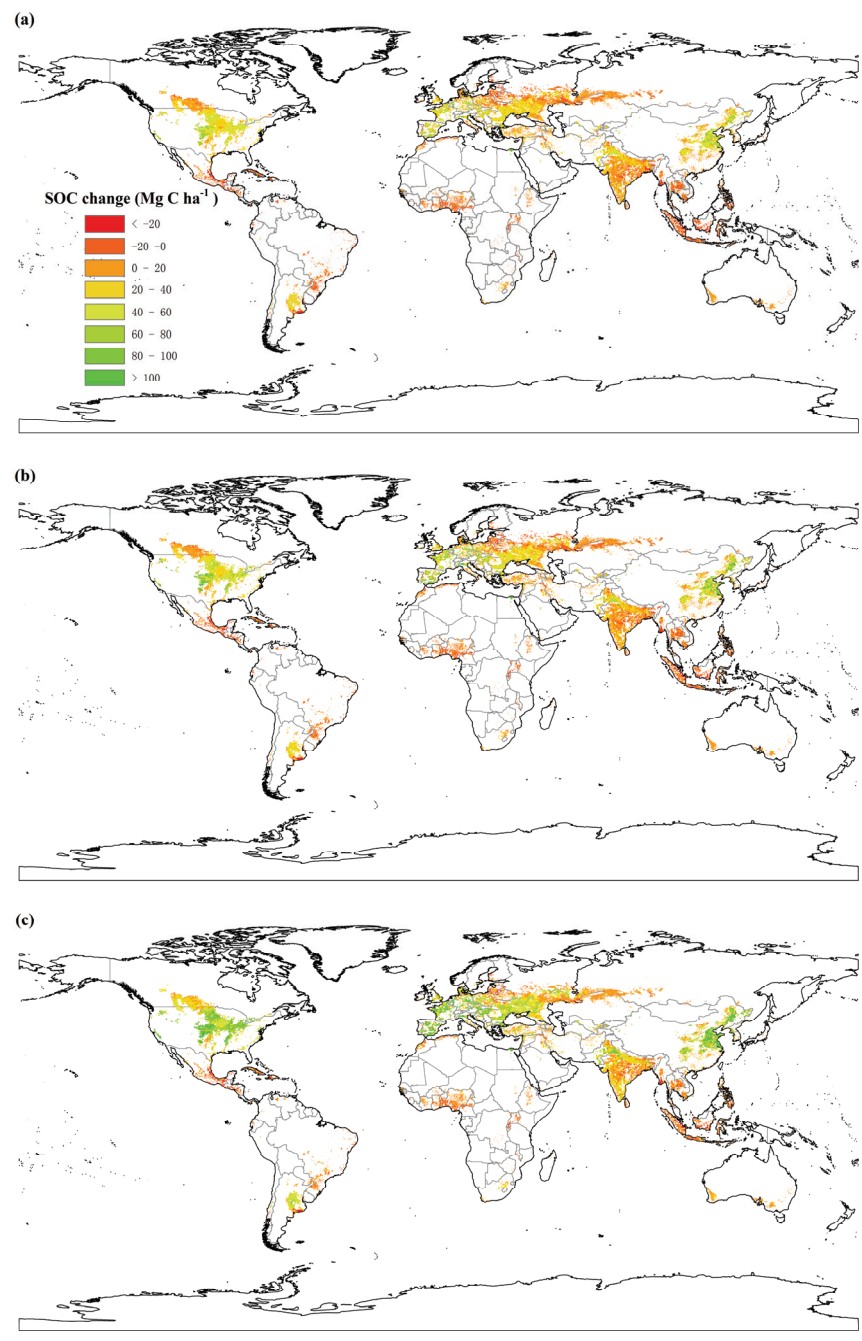

**Figure 2.** Spatial distribution of SOC change (1961-2014, MgC ha[-1]) across the global main cereal cropping regions under different above-ground crop residue retention rates of 30% (a), 60% (b) and 90% (c).





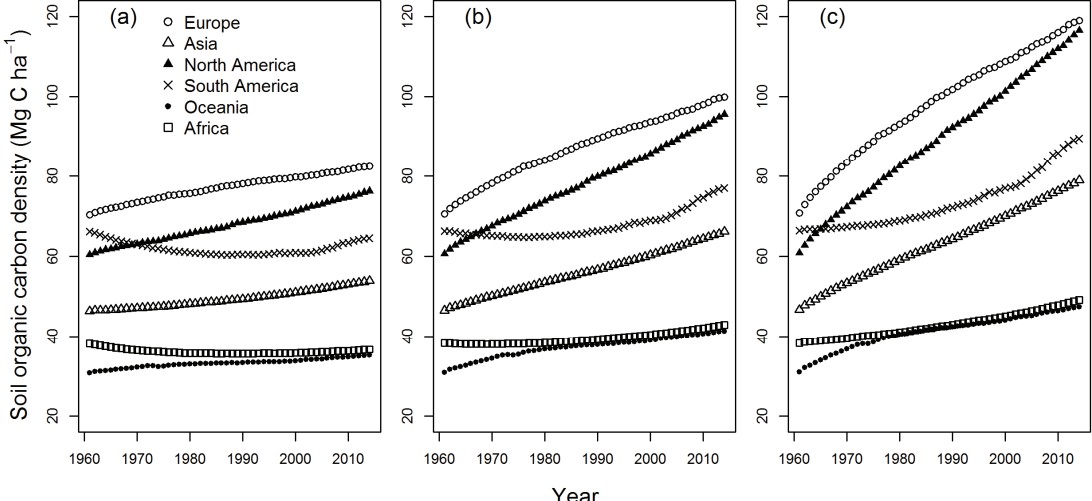

**Figure 3.** SOC evolution of five continents in the global main cereal cropping regions under different above-ground crop residue retention rates of 30% (a), 60% (b) and 90% (c).




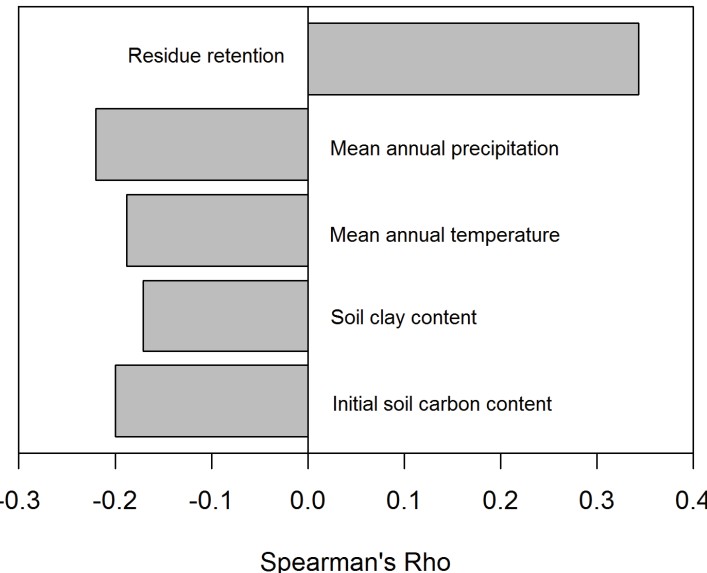

**Figure 4.** Spearman's rank correlation coefficients between SOC change (1961-2014, MgC ha$^{-1}$) and residue retention and soil and climate variables. All tests were significant ($P<0.001$).




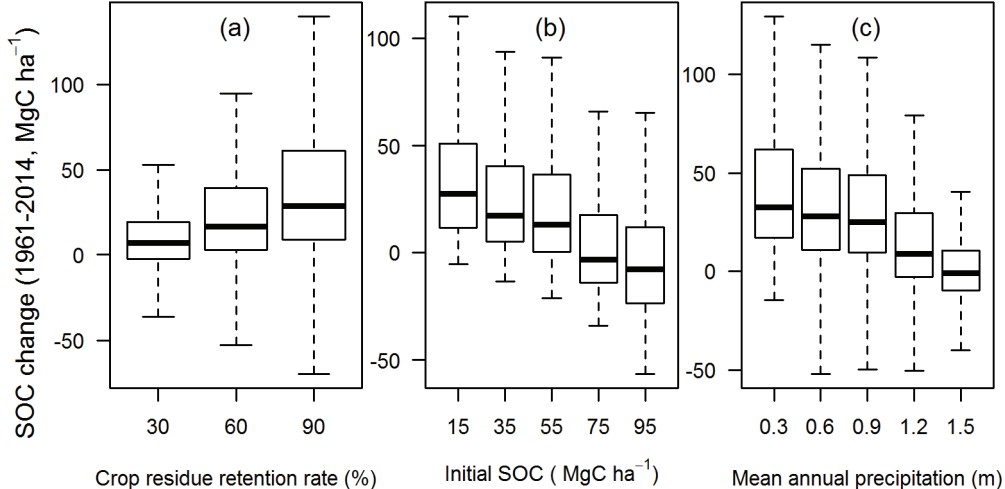

**Figure 5.** Response of SOC change (1961-2014, MgC ha$^{-1}$) to the three most influential variables of crop residue retention rate (a), initial SOC (b), and mean annual precipitation (c). Boxplots show the median and interquartile range, with whiskers extending to the most extreme data point within $1.5 \times (75\text{-}25\%)$ data range.