# Peer review of "Modeling soil organic carbon dynamics and their driving factors in the main global cereal cropping systems"

_Atmospheric Chemistry and Physics, 2017_

## Referee Comment (RC1) · Anonymous Referee #1 · 13 Jun 2017

The spatiotemporal variation of cropland soil organic carbon (SOC) across the global main cereal cropping system were analyzed based on the result of soil C turnover model (RothC), and the relationship between SOC changes and C input management, edaphic and climatic variables were also investigated in this article. Though, the modeled SOC may have bias due to the defect of the RothC model and the lack of model inputs datasets, it was still a better way to understand the spatiotemporal distribution of the SOC and its variation in a certain extent. It is very interesting and useful for the carbon input management and investigating the soil C sequestration on a global scale under the background of climate change. This paper is well presented, but the English of this paper need to be improved. The following is the concerns: 1.It was

confused by direct relationship between the net fluxes of carbon dioxide ($CO_2$) and soil organic carbon (SOC). "The net fluxes of carbon dioxide ($CO_2$) between the atmosphere and agricultural systems are mainly characterized by the changes in soil carbon stock, which. . ." in Page1 Line 10-12. $CO_2$ flux mainly depends on the $CO_2$ exchange between land surface and atmosphere by photosynthesis and respiration of the plant and decomposition of the microbe, but the variation of soil organic carbon was dominated by the carbon input. Detailed physical mechanism was suggested to be involved to link these two terms. And the same question is also found in Page 2 Line 8-9, "a small variation in soil carbon stock can lead to substantial changes in atmospheric carbon dioxide ($CO_2$) concentrations". 2.What does the abbreviation stand for? e.g. GIS, WISE, SOTER, HWSD, . . . 3.Why did the authors choose the 30%, 60% and 90% of the crop residue retention rates in this study? 4."enhancing the crop residue retention rate from 30% to 60% and 90% approximately induced a double and triple SOC sequestration rate, respectively (Fig. 1 and Fig. S3)" in Page 10 Line 20-21. It was difficulty to get the information of a double and triple SOC sequestration rate from these two Figures. 5. Because the air temperature and precipitation datasets are the input parameters, there should have some parameterization schemes to calculate the SOC based on the effect of temperature and precipitation in the RothC model. The derived SOC from the model has already included the information of climate change. How did you strip out this effect when attributing the variation of SOC under the background of climate change?

---

## Referee Comment (RC2) · Anonymous Referee #2 · 23 Jun 2017

This study simulated the spatiotemporal soil C dynamics across the global main cereal cropping systems using the RothC model and databases of soil and climate. The impacts of C input management, and soil and climatic variables on SOC changes were also analyzed. With the right reframing of the questions and additional detail, the study may become more novel and useful for the community. I think the study warrants publication in ACP after minor revision.

Detailed comments: 1. There is a focus on three crop residue retention rates (i.e., 30%, 60% and 90%) throughout the manuscript, yet the reason or context for this is not provided. 2. I suggest authors compare the present results with other modeling

studies for SOC changes at the global scale. 3. The modeled SOC density would be more valuable if the present results are compared with the observed SOC density in the five continents. 4. If a correction coefficient for RothC model be used to model SOC density in rice paddy, the results would be more reliable. I suggest authors discuss this issue by integrating corrected SOC density in rice paddy. 5. Change "cropland soil organic carbon" to "soil organic carbon in cropland".

---

## Author Response (AR1)

**G. Wang, on behalf of all authors (wanggc@mail.iap.ac.cn)**

**Reviewer #1:**

The spatiotemporal variation of cropland soil organic carbon (SOC) across the global main cereal cropping system were analyzed based on the result of soil C turnover model (RothC), and the relationship between SOC changes and C input management, edaphic and climatic variables were also investigated in this article. Though, the modeled SOC may have bias due to the defect of the RothC model and the lack of model inputs datasets, it was still a better way to understand the spatiotemporal distribution of the SOC and its variation in a certain extent. It is very interesting and useful for the carbon input management and investigating the soil C sequestration on a global scale under the background of climate change. This paper is well presented, but the English of this paper need to be improved.

**Authors' Response:** We greatly appreciate the reviewer's comments and their understanding of our work. We have submitted our MS to American Journal Experts (http://www.aje.com/) for editing and improvement of the English language.

The following is the concerns: 1.It was confused by direct relationship between the net fluxes of carbon dioxide (CO2) and soil organic carbon (SOC). "The net fluxes of carbon dioxide (CO2) between the atmosphere and agricultural systems are mainly characterized by the changes in soil carbon stock, which. . ." in Page1 Line 10-12. CO2 flux mainly depends on the CO2 exchange between land surface and atmosphere by photosynthesis and respiration of the plant and decomposition of the microbe, but the variation of soil organic carbon was dominated by the carbon input. Detailed physical mechanism was suggested to be involved to link these two terms. And , the same question is also found in Page 2 Line 8-9, "a small variation in soil carbon stock can lead to substantial changes in atmospheric carbon dioxide (CO2) concentrations".

**Authors' Response:** Yes, we have modified the sentences to the following form to avoid any possible misunderstandings:

"Changes in the soil organic carbon (SOC) stock are determined by the balance between the carbon input from organic materials and the output from soil C decomposition. The fate of SOC in cropland soils plays a significant role in both sustainable agricultural production and climate change mitigation."

For the second point, we further clarified this in the revised MS: "On a global scale, the soil is the largest terrestrial carbon pool, and it stores approximately three times the quantity of C that is in the atmosphere. Consequently, a small variation in soil carbon stock can lead to substantial changes in atmospheric carbon dioxide (CO2) concentrations". This is a widely accepted view in the existing literature, please refer to Cleveland and Townsend (2006), Davidson and Janssens (2006), Luo *et al.* (2010), and West and Post (2002).

2. What does the abbreviation stand for? e.g., GIS, WISE, SOTER, HWSD, ...

**Authors' Response:** We have clarified these abbreviations in the revised MS. GIS: geographic information system; WISE: World Inventory of Soil Emission Potentials; SOTER: Soil Terrain Database; HWSD: Harmonized World Soil Database.

3. Why did the authors choose the 30%, 60% and 90% of the crop residue retention rates in this study?

Authors' Response: We have explained and clarified the use of these rates in the revised MS (XXX): "The crop residue that is retained in the system after harvest can benefit the sequestration of soil carbon in the croplands. The amount of above-ground residue that is retained in the system, however, shows vast spatial disparity and uncertainty across the global croplands. In developing regions such as Asia and Africa, it has been suggested that only approximately 30% of the crop residues are retained in the soils after harvest (Jiang et al., 2012; Baudron et al., 2014). In developed regions such as Europe and North America, however, the crop residue retention rate can reach over 60% (Scarlat et al., 2010; Lokupitiya et al., 2012). Furthermore, in Australia, it has been reported that 100% of the crop residue was retained across 72–100% of the cropping area of the country from 2010 to 2014 (National Inventory Report, 2013, 2015). However, this information is based on rough estimations and statistical data. To the best of our knowledge, detailed information on the residue retention rates over a meaningfully large scale of both time and space across different countries and continents is still lacking. Consequently, a scenario modeling approach was adopted to assess the dynamics of SOC as determined by various potential management practices on crop residues. We specified three crop residue retention rates in the present study, i.e., 30%, 60% and 90%."

These three scenarios represent the residue retention rates typically adopted in developing regions with relatively poorly managed systems (30%), developed regions with better managed systems (60%), and the areas with well-managed agricultural conservation systems (90%).

4."enhancing the crop residue retention rate from 30% to 60% and 90% approximately induced a double and triple SOC sequestration rate, respectively (Fig. 1 and Fig. S3)" in Page 10 Line 20-21. It was difficulty to get the information of a double and triple SOC sequestration rate from these two Figures.

Authors' Response: We have clarified this in the revised MS:

"On a global average, the total amount of C input to soils is 1.7, 2.7 and 3.7 Mg C ha-1 under the crop residue retention rates of 30%, 60% and 90%, respectively (Fig. S3). The corresponding annual rates of SOC changes under R30, R60 and R90 were 0.22, 0.45 and 0.69 Mg C ha-1 yr-1, respectively (Fig. 1), indicating approximately doubled and tripled SOC sequestration rates after enhancing the residue retention rate from 30% to 60% and 90%."

5. Because the air temperature and precipitation datasets are the input parameters, there should have some parameterization schemes to calculate the SOC based on the effect of temperature and precipitation in the RothC model. The derived SOC from the model has already included the information of climate change. How did you strip out this effect when attributing the variation of SOC under the background of climate change?

**Authors' Response:** The quantified dynamics of soil carbon are regulated by complex interactions between C input, climate conditions such as temperature and precipitation, and soil conditions such as initial soil C density and clay fraction. To assess the contribution of each controlling factor to soil C changes, we adopted the Spearman's rank correlation approach, using the *cor.test* function in the *stats* package in R. The sign of Spearman's rank correlation coefficient (rho), positive or negative, indicates the direction of the association between the independent and dependent variables. The absolute magnitude of rho, between 0 and 1, suggests the strength of the correlation between the two variables. We have specified this approach in the Methods section and presented the statistical analysis results in the Results section (and Fig. 4).

**Reviewer #2:**

This study simulated the spatiotemporal soil C dynamics across the global main cereal cropping systems using the RothC model and databases of soil and climate. The impacts of C input management, and soil and climatic variables on SOC changes were also analyzed. With the right reframing of the questions and additional detail, the study may become more novel and useful for the community. I think the study warrants publication in ACP after minor revision.

Authors' Response: We greatly thank the reviewer for their thoughtful comments and understanding of our work.

Detailed comments: 1. There is a focus on three crop residue retention rates (i.e., 30%, 60% and 90%) throughout the manuscript, yet the reason or context for this is not provided.

Authors' Response: As mentioned above, we have provided more information and clarified this in the revised MS: "The crop residue that is retained in the system after harvest can benefit the sequestration of soil carbon in the croplands. The amount of above-ground residue that is retained in the system, however, shows vast spatial disparity and uncertainty across the global croplands. In developing regions such as Asia and Africa, it has been suggested that only approximately 30% of the crop residues are retained in the soils after harvest (Jiang et al., 2012; Baudron et al., 2014). In developed regions such as Europe and North America, however, the crop residue retention rate can reach over 60% (Scarlat et al., 2010; Lokupitiya et al., 2012). Furthermore, in Australia, it has been reported that 100% of the crop residue was retained across 72–100% of the cropping area of the country from 2010 to 2014 (National Inventory Report, 2013, 2015). However, this information is based on rough estimations and/or statistical data. To the best of our knowledge, detailed information on the residue retention rates over a meaningfully large scale of both time and space across different countries and continents is still lacking. Consequently, a scenario modeling approach was adopted to assess the dynamics of SOC as determined by various potential management practices on crop residues. We specified three crop residue retention rates in the present study, i.e., 30%, 60% and 90%."

These three scenarios represent the residue retention rates typically adopted in developing regions with relatively poorly managed systems (30%), developed regions with better managed systems (60%), and the areas with well-managed agricultural conservation systems (90%).

2. I suggest authors compare the present results with other modeling studies for SOC changes at the global scale.

**Authors' Response:** We have further compared the global cropland soil C sequestration rates quantified in this study to the estimations of Lal (2004). The efficiency of the conversion of C input to SOC (i.e., ratio of SOC change to C input) estimated in the present study was compared to that of Campbell *et al.* (2000) in the revised MS. We found that our modeled results are comparable and fall within the ranges of their estimations.

3. The modeled SOC density would be more valuable if the present results are compared with the observed SOC density in the five continents.

**Authors' Response:** In this study, we adopted the HWSD soil dataset (can be referred to as the observed SOC density) as one of the model's driving inputs, and our goal was to simulate the soil carbon changes under changing environmental and management conditions during the last half century. As such, the modeled SOC density in the final year is highly dependent on the initial SOC density (HWSD soil dataset, also as the model's soil input data) and the modeled SOC changes. Comparing the soil C changes to the initial SOC density (observed SOC density) is meaningful and useful to extrapolate the regulating effects of soil conditions on SOC dynamics. We assessed the impacts of initial SOC density on the modeled SOC changes in the present study (Fig. 4 and Fig. 5), and found that under otherwise similar conditions, the soil would lose more C with a higher initial SOC density, and *vice versa*.

4. If a correction coefficient for RothC model be used to model SOC density in rice paddy, the results would be more reliable. I suggest authors discuss this issue by integrating corrected SOC density in rice paddy.

Authors' Response: We have discussed this issue in the revised MS:

"Second, the RothC model was developed to simulate the soil organic matter turnover in upland soils (Jenkinson et al., 1990), and it generally performs well in the global wheat systems with non-waterlogged soils (Wang et al., 2016). In the paddy soils, particularly during the rice-growing seasons, the soil C decomposition rate might be reduced when subjected to anaerobic conditions. For example, Shirato and Yokozawa (2005) used the RothC model to simulate the C changes in Japanese paddy soils and suggested that the model's performance can be improved by modifying the SOC decomposition rates during the rice growing-season. As such, the default parameters adopted in the present study may bias the simulations of the SOC changes across the rice systems are that mainly distributed in the Southeast Asia. In the present study, we adopted the model's default parameters rather than the modified factors from Shirato and Yokozawa (2005) mainly because the rice-growing areas in Japan constitute approximately 1% of the world's total (FAOSTAT, 2017), and the associated climatic and edaphic conditions differ significantly from the other rice systems. We highlight the need to robustly calibrate the model's soil C decomposition rates against the long-term experimental data across the rice paddy soils to represent the different patterns in climate, soil and management conditions of the Southeast Asia in the future."

5. Change "cropland soil organic carbon" to "soil organic carbon in cropland".

Authors' Response: Modified accordingly.

The net fluxes of carbon dioxide (CO2) between Changes in the atmosphere and agricultural systems are mainly characterized by the changes in soil organic carbon (SOC) stock, which is are determined by the balance between the carbon input from organic materials and the output through soil C from the decomposition of soil C. The fate of SOC in cropland soils plays a significant role in both sustainable agricultural production and climate change mitigation. The spatiotemporal changes of cropland soil organic carbon (SOC) in croplands in response to different carbon (C) input management and environmental conditions across the main global-main cereal systems were studied using a modeling approach. We also identified the key variables drivingthat drive SOC changes at a high spatial resolution ( $0.1^{\circ} \times 0.1^{\circ}$ ) and over a long time scale (54 years from 1961 to 2014). TheA widely used soil C turnover model (RothC) and the state-of-the-art databases of soil and climate variables were used in the present study. The model simulations suggested that, on a global average, the cropland SOC density increased at an annual rate of 0.22, 0.45 and 0.69 MgC ha-1 yr-1

under a-crop residue retention raterates of 30%, 60% and 90%, respectively. IncreasedIncreasing the quantity of C input could enhance the soil C sequestration or reduce the rate of soil C loss-rate, depending largely on the local soil and climate conditions. Spatially, under a certainspecific crop residue retention rate, a-relatively higher soil C sinksinks were-generally found across the central parts of the United States, western Europe, and the northern regions of China, while a relatively. Relatively smaller soil C sink generallysinks occurred in the high latitude regions at high latitudes of both the northern and southern hemispherehemispheres, 
[revised manuscript text omitted]
 contentcontents generally had a-higher clay fraction fractions, and this would overshade overshadow the beneficial contributions of soil clay in benefiting to soil C accumulation.

5

10

15

The negative effects of higher temperature and precipitation on SOC change identified in the present study (Fig.Figs. 4 and 5) can be attributed to the higher SOC decomposition raterates in warmer and wetter soils, which is consistent with the RothC model's description of the RothC model (Jenkinson et al., 1990) and the other findings of by Bond-Lamberty and Thomson (2010). Here, it should be noted that such correlations between the climate and SOC ehangechanges might only be valid in a soil carbon turnover model consisting that only consists of the-C dynamic C processes in the soil (e.g., RothC model). In other agricultural system-model simulations, climatic variables may play a different role in affecting the SOC change through jointly regulating both crop productions and soil C dynamics. For example, Wang et al. (2014) used a process-based agricultural system model (i.e., Agro-C model) to simulate the SOC dynamics in the semi-arid regions of the North China Plain, and found a-positive effects of temperature and precipitation on SOC accumulation. This is because, in the-temperature and water deficient areas (e.g., the\_North China Plain), increased temperature and precipitation promoted promote crop production and hence increasing increases the C input to soils-and favoring, which favors SOC sequestration.

Can we estimate the actual historical soil C dynamics across the world? There exists a bigA large challenge exists due mainly to a lack of data availability, particularly for the two main RothC model inputs such as initial SOC content and annual C input. FirstlyFirst, the soil properties presented withby the HWSD were derived from different sources with unevenunevenly sampled soil profiles over time and space. As such, the value of initial SOC content can hardly, if not impossibleat all, represent the actual SOC content inat the beginning of the study period. However, the modeled dynamics of the\_SOC in the present study eanmay be validappropriate, to a certain extent, to represent the spatiotemporal patterns of the soil C source/\_and\_sink processes. Second, there remains a lack of detailed information on crop residue management across both time and space\_remains, which\_also hinders our ability to accurately characterize the SOC changes on a large scale at fine spatiotemporal resolutions. Such as it isHowever, we can still roughly assume that the above-ground residue retention rates were generallyapproximately 30% in developing regions such as Asia and Africa (Jiang et al., 2012; Baudron et al., 2014; Erenstein, 2011);) and 60% in other regions (Lokupitiya et al., 2012; Scarlat et al., 2010; Baudron et al., 2015). Based on these assumptions, we furtheredfurther

quantified that the global average SOC increased at a rate of 0.34 MgC ha-1 yr-1 atunder an average annual C input rate of 2.4 MgC ha-1 yr-1 from 1961 to 2014. On a global scale, the estimated efficiency of the conversion of C input <del>C</del>-to SOC (i.e., the ratio of SOC change to C input) equals to is 14%, which falls intowithin the 10-18% range of estimated by Campbell et al. (2000)'s result of 10-18%.). It should be noted that the conversion efficiency varies across space and is highly dependent on the local climatic and edaphic conditions (Yu et al., 2012).\_

5

By extrapolating these results to the world's whole croplands with a global total cropland area of 1,400 Mha (Jobbagy and Jackson, 2000), it was found that the global cropland soils could have annually sequestered 0.48 Pg C annually from 1961 to 2014, which equals to around approximately 8% of the contemporaneous global average annual C emissions from fossil fuel combustions (http://cdiac.ornl.gov/ftp/ndp030/global.1751\_2014.ems). ThroughBy enhancing the crop residue retention raterates to 60% and 90% in all the global croplands, the soil C accumulation would offset around approximately 11% and 16%, respectively, of the fossil fuel-induced C emissions. Again, it is noteworthy that although Although soil C can be increased by enhancing the quantity of C input, it would eventually reach a threshold at a higher level (Stewart et al., 2007). Until then, more amount of carbon input would be needed to maintain the soil C at higher levels (Wang et al., 2016). Otherwise, the cropland-soil C in croplands would decrease, and soils would again act as a net C source again.

10

**4.2 Uncertainties and limitations**

Several uncertainties and limitations should be noticed inconsidered when interpreting the simulation results in this study. First, the modeled SOC change modeled in the present study could be biased due to the spatial inconsistency in the time of soil sampling, which generally rangesvaried widely duringover the second half of the twentieth century (Fao and Isric, 2012). In some places, the initial soil C information derived from the HWSD could only represent the actual soil C levels during the later periods other than after the early 1960s-. For example, the soil profile measurements used for producing produce the soil map of China, –which is further-included in the HWSD datasets, were generally madecollected in the late 1970s and early 1980s (Yu et al., 2007). Considering that the

spatial patterns of cropland SOC-across space could have substantially changed over the study period under the changing environments and management practices (FigFigs. 1 and 2), the initial SOC used in the present study (derived from HWSD) might significantly differ from the actual soil C levels in the early 1960s. BesidesIn addition, it has been reported that soils with higher initial C contents would experience smaller increase or greater C losslosses under otherwise similar conditions, and vice versa (Zhao et al., 2013; Wang et al., 2015). Consequently, for those regions with soil sampling timetimes much later than the early 1960s, our quantified SOC changes could may be underestimated underestimations in the areas with where substantial soil C increase increases had occurred before then. On the contrary measurements were collected. In contrast, the SOC changes could be overestimated acrossin the areas accompanying with that are accompanied by a previous significantly decreased significant decrease in soil C.

10 Second, the RothC model was developed for simulating to simulate the soil organic matter turnover in upland soils (Jenkinson et al., 1990), and it generally showed performs well performance acrossin the global wheat systems with non-waterlogged soils (Wang et al., 2016). In the paddy soils, particularly during the rice-growing seasons, the soil C decomposition rate might be reduced when subjected to anaerobic conditions. For example, Shirato and Yokozawa-(2005). Consequently.) used the RothC model, used to simulate the C changes in Japanese paddy soils and suggested 15 that the model's performance can be improved by modifying the SOC decomposition rates during the rice growing season. As such, the default parameters adopted in the present study, could have underestimated may bias the simulations of the SOC changes across the rice systems that are mainly distributed in the Southeast Asia. In the present study, we adopted the model's default parameters rather than the modified factors from Shirato and Yokozawa (2005) mainly because the rice-growing areas in Japan constitute approximately 1% of the world's total (FAOSTAT, 2017). 20 and the associated climatic and edaphic conditions differ significantly from the other rice systems. We highlight the need to robustly calibrate the model's soil C decomposition rates against the long-term experimental data across the rice paddy soils to represent the different patterns in climate, soil and management conditions of Southeast Asia in the future.

Last but not leastFinally, the limitations of the current first-order decay model (e.g., RothC) may cause significant bias in the model simulations. For example, our results suggested a general linear relation between C input and SOC variation (FigFigs. 1 and S3), which contradicts theprevious findings ofthat increasing the incorporated amount of crop residue incorporation may affect the SOC change in variablea variety of ways other than linearly (Powlson et al., 2011). Moreover, it has been reported that although soil can accumulate a significant amount of C when the pre-existingpreexisting soil C content is low, the SOC reaches a certain higherthreshold level (i.e., carbon saturation state) withwhere little or no significant further changechanges occur even withwhen more C inputsis added (Stewart et al., 2007; Qin et al., 2013). Without considering the C saturation state, the first-order decay model might overestimate the SOC atin longer time scale simulations particularly in regions withwhere the C input is higher C-input and lowerthe 
[revised manuscript text omitted]